# CSF Metabolomics of Tuberculous Meningitis: A Review

**DOI:** 10.3390/metabo11100661

**Published:** 2021-09-28

**Authors:** Shayne Mason, Regan Solomons

**Affiliations:** 1Human Metabolomics, Faculty of Natural and Agricultural Sciences, North-West University, Potchefstroom 2531, South Africa; 2Department of Paediatrics and Child Health, Faculty of Medicine and Health Sciences, Stellenbosch University, Tygerberg 7505, South Africa; regan@sun.ac.za

**Keywords:** cerebrospinal fluid (CSF), tuberculous meningitis (TBM), metabolomics, metabolic characterization

## Abstract

From the World Health Organization’s global TB report for 2020, it is estimated that in 2019 at least 80,000 children (a particularly vulnerable population) developed tuberculous meningitis (TBM)—an invariably fatal disease if untreated—although this is likely an underestimate. As our latest technologies have evolved—with the unprecedented development of the various “omics” disciplines—a mountain of new data on infectious diseases have been created. However, our knowledge and understanding of infectious diseases are still trying to keep pace. Metabolites offer much biological information, but the insights they permit can be difficult to derive. This review summarizes current metabolomics studies on cerebrospinal fluid (CSF) from TBM cases and collates the metabolic data reported. Collectively, CSF metabolomics studies have identified five classes of metabolites that characterize TBM: amino acids, organic acids, nucleotides, carbohydrates, and “other”. Taken holistically, the information given in this review serves to promote the mechanistic action of hypothesis generation that will drive and direct future studies on TBM.

## 1. Introduction

Tuberculosis (TB) is an age-old disease in humans that needs no introduction. The bacterium responsible for the condition—*Mycobacterium tuberculosis* (*M. tb*)—has infected humankind for millennia; it was identified in Levantine populations during the pre-pottery Neolithic C period, more than 8000 years ago [1]. The World Health Organization (WHO)’s global TB report for 2020 concluded that approximately 10 million people contracted the disease in 2019 [2]: 56% were men, 32% were women, and 12% were children (aged < 15 years). Eight countries accounted for more than two-thirds of these cases—India (26%), Indonesia (8.5%), China (8.4%), the Philippines (6.0%), Pakistan (5.7%), Nigeria (4.4%), Bangladesh (3.6%), and South Africa (3.6%). The geographical regions considered most burdened with TB are located almost exclusively within South-East Asia and sub-Saharan Africa (Figure 1).

Tuberculous meningitis (TBM) is the infection of the meninges by *M. tb*. The exact mechanism of how the bacterium invades the central nervous system is still debated. However, what is certain, and well documented in the literature, is that advanced TBM presents with very severe clinical symptoms, and is often considered the most severe manifestation of TB. Untreated, TBM is invariably fatal, and even if treated, mortality and morbidity are high, particularly in children [3]. It has been reported that in a high-TB setting approximately 10% of hospital-based TB cases in children have TBM [4]. Thus, if 12% of new TB infections occurred in children globally in 2019, and two-thirds of incidences arose in high-TB-burden regions [2], then it is postulated that at least 80,000 children developed TBM in 2019. However, the estimate of the global burden of TBM is complicated by many cases that are undiagnosed, and many treated cases are believed to be unreported [3]. Hence, 80,000 cases of TBM in children in 2019 is likely an underestimate.

A global network of experts in TBM—the Tuberculous Meningitis International Research Consortium (TMIRC)—have acknowledged that there are many facets of TBM that require greater understanding. The consortium convened for the first time in Cape Town, South Africa, in 2009 to discuss all things related to TBM, and the way forward. Since 2009, the TMIRC has met in Dalat, Vietnam, in 2015; in Lucknow, India, in 2019; and was hosted (virtually) by TBM experts in Bandung, Indonesia, in 2020. These meetings have resulted in a resurgence of research on TBM, as evident by the growing number of corresponding peer-reviewed scientific publications (Figure 2). The TMIRC co-authored 15 papers between 2016 and 2020, in which historical and current research on TBM was reviewed, important gaps in knowledge identified, and research priorities defined [3,5,6,7,8,9,10,11,12,13,14,15,16,17,18]. A clear gap in knowledge about TBM is the paucity of metabolic characterization of the disease.

Metabolic characterization here refers to explaining TBM in terms of disrupted metabolic pathways and up/downregulated metabolites. Within the central dogma of biochemistry, DNA in cells sends out genetic information via transcription to mRNA, which then translates this information to generate proteins. These proteins, via hormones and enzymatic reactions, then regulate metabolism. Metabolites—low-molecular-weight compounds—are the end products of these processes, and contain a vast amount of biological information, expressing great physicochemical complexity. Metabolomics—the science of examining low-molecular-weight compounds in biological systems—has allowed us to scratch the surface of this iceberg of metabolic information about TBM. This review identifies and provides an overview of the CSF metabolomics studies in the literature that provide metabolic information that may help to characterize the disease. This consolidated metabolic information is aimed at promoting the mechanistic action of hypothesis generation to drive future TBM research.

## 2. CSF Metabolomics of TBM

### 2.1. Literature Search Parameters

Using the online search criteria of “tuberculous meningitis” and “metabolomics” in PubMed, up to the end of 2020, yielded 14 hits; of these, 11 studies [19,20,21,22,23,24,25,26,27,28,29] contained metabolic information on TBM in CSF from humans, and are discussed in this review. Further examination of these 11 studies yielded three more CSF metabolomics studies on TBM [30,31,32]. A further online search in PubMed, using the search criteria “tuberculous meningitis” and “metabolism” in the title and/or abstract, from 1990 to 2020, yielded 18 hits. All of these studies were examined, but only four [33,34,35,36] contained metabolic information on TBM in CSF from humans, and are also included in this review. Further inspection of these sources’ references identified eight additional studies containing metabolic information on TBM in CSF from humans [37,38,39,40,41,42,43,44]. Of note, PubMed shows that numerous biochemical studies from the 1950s and 1960s were published on the metabolism/metabolic aspects of TBM; however, the full texts of these publications are not available online, and most of them are not in English. Furthermore, lipidomics (lipid-targeted metabolomics) was not included in this review.

### 2.2. Metabolomics—The Basics

The term “metabolome” was first used in 1998 by Oliver et al. [45], and the actual term “metabolomics” was first used by Oliver Fiehn in 2001 [46]. Metabolomics is a multidisciplinary scientific field that incorporates the sciences of chemistry, biology, and statistics, along with biochemistry, biostatistics, and chemometrics. The principal driver behind metabolomics has been the use of highly sophisticated analytical instruments that generate data-rich outputs reflecting the metabolism of biological organisms and/or samples. The two key platforms employed within metabolomics are nuclear magnetic resonance (NMR) spectroscopy and mass spectrometry (MS), typically coupled with a form of chromatographic separation—for example, gas chromatography (GC) or liquid chromatography (LC). The scientific approach in metabolomics can either be an open-minded, hypothesis-generating one in which a global view of the metabolome is examined (called the untargeted approach), which offers descriptions of discoveries, or one in which a specific subset of the metabolome is examined with an a priori, hypothesis-driven agenda (the targeted approach) that gives mechanistic insights. The application of metabolomics to TBM specifically has been addressed in past reviews—one focused on NMR [47], and another on MS [48], both of which advocated that metabolomics will yield data that improve existing knowledge of the disease. This review consolidates the CSF metabolomics data on TBM collected in the past 15 years.

### 2.3. History of CSF Metabolomics Studies of TBM in Human Subjects

One of the first metabolomics studies involving CSF collected from humans with TBM was a 2005 pilot investigation [30] that compared various forms of meningitis cases to controls in order to determine the feasibility of CSF metabolomics as a technique for the rapid diagnosis of meningitis. The etiological agents of the 23 meningitis cases analyzed by Coen et al. [30] were fungal (*n* = 1), viral (*n* = 11), and bacterial (*n* = 9); only two cases were diagnosed as TBM. Unfortunately, there were insufficient TBM subjects available in this study to characterize the disease. However, the significance of this work was that it was able to differentiate viral meningitis from other forms (fungal, bacterial, and tubercular) of meningitis, based solely on NMR metabolomics. Furthermore, Coen et al. [30] laid the foundation for future CSF metabolomics studies on TBM by reporting that NMR metabolomics is rapid, requires minimal sample preparation, and can be considered as a potential diagnostic platform used for the differentiation of bacterial from viral meningitis.

The next CSF metabolomics study involving TBM appeared a decade later, in 2015, and was the first human study under the umbrella of metabolomics that focused on TBM [22]; this study, by Mason et al. [22], was an untargeted NMR-based metabolomics investigation using CSF from a pediatric (<12 years old) cohort from a South African population. In the original cohort, 33 TBM cases were compared to 30 “sick” and 43 “healthy” age-matched controls. After outlier removal, in order to make each group as homogeneous as possible, 17 TBM cases were compared to 19 “sick” and 30 “healthy” controls by means of various statistical analyses. The most dominant differentiating CSF metabolites were reduced glucose (2.69 ± 1.22 mmol/L) and highly elevated lactate (7.36 ± 2.36 mmol/L) in TBM cases (see Table 1). An additional 15 CSF metabolites were identified as characterizing TBM—namely, decreased acetate, citrate, myo-inositol, creatinine, and dimethyl sulfone, and increased alanine, lysine, isoleucine/leucine, valine, phenylalanine, tyrosine, pyruvate, formate, and choline. The significance of this study was that it yielded the first metabolic picture of TBM based on CSF metabolomics. The 15 metabolites characteristic of TBM indicated perturbed neuroenergetics. As a consequence, a link between astrocytes and microglia was proposed in terms of a hypothesis expressed as the “astrocyte–microglia lactate shuttle” (AMLS). Briefly, this hypothesis states that complex cascading communications involving energy-associated metabolites—from the astrocytes to the microglia—are needed by the microglia-driven neuroinflammatory response to the invading *M. tb*. Another feature was that the metabolic characterization was conducted using two control groups—one “sick” (the same cohort but confirmed negative for any form of meningitis, despite the subjects being ill with symptoms reminiscent of the condition), and one “healthy” (a different cohort, suspected of suffering from a neurometabolic disease). After detailed investigations, however, no clinical or biochemical evidence was found for diagnosis of TBM in any of these patients. The study by Mason et al. [22] had two major limitations: (1) It is important to note, reflecting a common limitation across many CSF studies, that it is not ethical to obtain CSF samples from healthy individuals for research purposes, especially pediatric cases, because the collection of CSF via lumbar puncture carries inherent health risks. This limitation was partially overcome by using an additional (“healthy”) control group of patients who were considered to be “normal” based on their CSF metabolic profiles. (2) Unassigned variables (NMR spectral regions that could not be assigned a metabolite name) were identified as being statistically significant, but were not investigated further, because these unassigned variables held no biological value—that is, they were not associated with a specific metabolite, so they could not be discussed in terms of metabolism. Hence, potentially important biological and/or diagnostic information was not assessed.

The following year (2016), another untargeted NMR-based metabolomics study was conducted on an adult (>18 years old) Indian cohort. Chatterji et al. [31] examined the CSF, serum, and urine from TBM cases, and compared them with controls and bacterial meningitis (BM) cases. Of the 83 CSF samples analyzed, 30 represented TBM, 21 BM, and 22 controls. Chatterji et al. employed a typical untargeted metabolomics approach on the binned NMR spectral data; however, unsupervised principal component analysis did not differentiate TBM cases from controls, nor from BM subjects. The reasoning given was that binning possibly compromised the spectral resolution of the dataset. Chatterji et al. [31] used an in-house software protocol called QUANTAS (for QUANTification by Artificial Signal), whereby 26 metabolites were first identified and quantified from the NMR spectra, and then assessed via univariate statistics, discriminant function analysis and, lastly, multivariate statistics. This approach to the metabolomics data was unique, but 15 CSF metabolites were identified that characterized TBM. Reduced glucose and elevated lactate were the two predominantly differentiating metabolites. The 13 other metabolites were decreased myo-inositol, acetate, and 3-hydroxyisovalerate, and increased isoleucine, valine, histidine, phenylalanine, tyrosine, isobutyrate, 2-hydroxybutyrate, creatine, glycerophosphocholine, and formate (Table 1). These 15 CSF metabolites were attributed to infection and neuroinflammation. Of them, a combination of four metabolites (decreased glucose, and increased lactate, pyruvate, and citrate) was identified as having the potential to correctly classify TBM. Similarly, a combination of three metabolites (decreased 3-hydroxyisovalerate, isobutyrate, and formate) was used to distinguish TBM from BM. The significance of the report by Chatterji et al. [31] is that it was the first metabolomics study to characterize the metabolic profiles of three different biofluids from adult TBM cases, and to differentiate between TBM and BM. It was also the first metabolomics study to suggest a potential diagnostic CSF biosignature of TBM, with a sensitivity and specificity of 73.3% and 80.7%, respectively. The major limitations were that the HIV status of the participants was unknown and the metabolic characterization of TBM was performed by comparing TBM cases with neurological controls—cases proven negative for meningitis, but with neurological symptoms, such as migraine, microcytic anemia, lumbar arachnoiditis, cranial nerve palsy, paraparesis, and hemiparesis. Furthermore, relatively few common metabolites were identified across the three biofluids analyzed. Chatterji et al. later published a similar metabolomics study in 2017 [32]; however, all metabolites representative of meningitis were placed in a single experimental group; thus, none of the results can be attributed directly to TBM.

In 2017, an untargeted NMR metabolomics study was conducted on an adult (15–70 years old) Chinese population that compared CSF samples (N = 38) between TBM (*n* = 18) and viral meningitis (VM; *n* = 20) cases [21]. The major limitation here, with respect to this review, was that no comparison was made with control cases; hence, TBM could not be characterized. Another limitation was that only the NMR spectral region in the range 0.4–4.4 ppm was analyzed. Thus, all aromatic chemical compounds (for example, phenylalanine and tyrosine) in the NMR spectral region between 7.0–9.0 ppm were not considered. Li et al. [21] identified 25 CSF metabolites that differentiated TBM from VM—14 (56%) increased, and 11 (44%) decreased in TBM. Of these 25 metabolites, 4 discriminating metabolites (glycine, tyrosine, glutamine, and serine) were identified as being potential biomarkers for differential diagnosis of TBM from VM, based on the statistical cutoff criteria of VIP > 2 and FC > 2 or <0.5. The significance of this study, as stated by Li et al. [21], is that the significant change in carbohydrate (and lipid) metabolism in TBM cases indicates increased neuroenergetics and anaerobic metabolism, which is most likely associated with the invading *M. tb* and the host neuroinflammatory response. Another strength is that this was the first metabolomics study on TBM that employed pathway analysis using MetaboAnalyst 3.0 (www.metaboanalyst.ca/ (accessed on 20 September 2021)) to highlight the metabolic pathways most affected by TBM. One of the major limitations was that many metabolites were reported at a low intensity. NMR is known to have much lower sensitivity than its MS-based analytical counterparts; however, based on the CSF sample preparation method described by Li et al. [21], the CSF samples were diluted 1.83 times, confounding the issue of low sensitivity of NMR. Other limitations were that (1) lipoprotein was identified as significant, but lipoproteins should not be present in the samples, since 0.22 μmol/L centrifugal filters were used; (2) malonic acid and malonate—the same metabolites—were both identified independently as being significant; and (3) cyclohexane and N,N-dimethylformamide were identified as being statistically significant, but the biological origin of these chemical compounds is not known.

The year 2017 also saw the first GC–MS-based metabolomics studies on TBM. Using a targeted GC–MS approach, Mason et al. [25] profiled the amino acids in CSF collected from a South African pediatric cohort (<12 years old), consisting of 33 TBM cases and 34 controls. A total of 29 amino acids were measured, but were accepted only via a stringent quality control procedure. Hence, any variation detected in these 15 amino acids can be attributed to biological variation (as should be the focus of any metabolomics study), with minimal (negligible) variation from the analytical method. Herein lies the strength of this targeted metabolomics study [25]—the strict quality control procedures removed any amino acids that exhibited too much analytical (unwanted) variation from the analytical method. From the quality control assessment, five amino acids indicated good reliability and, hence, were confidently identified as the important discriminatory variables. These five amino acids were alanine (TBM: 77.3 ± 52 μmol/L, control: 25 ± 9.5 μmol/L), asparagine (TBM: 19.9 ± 14 μmol/L, control: 5.3 ± 1.7 μmol/L), glycine (TBM: 58.2 ± 31.7 μmol/L, control: 24.4 ± 1.4 μmol/L), lysine (TBM: 36.5 ± 25.2 μmol/L, control: 14.5 ± 5.2 μmol/L), and proline (TBM: 24.3 ± 25.8 μmol/L, control: 1.8 ± 0.5 μmol/L). Mason et al. [25] provided some insights into these five amino acids that characterize TBM; they proposed a link to neuroenergetics (alanine and asparagine), neurotransmitter inhibition (alanine and glycine), a connection with lipid peroxidation (lysine), a link to the glutamine–glutamate cycle (proline and asparagine), and coincidence with elevated CSF protein (proline). The common factors with these five amino acids were ammonia metabolism and removal of excess nitrites [51].

Two LC–MS-based metabolomics studies have also been performed using CSF samples from human TBM cases. For example, Dai et al. [19] examined 50 TBM cases in a Chinese adult (>18 years old) population, and provided extensive statistical output; however, no comparison was made with a control group—that is, TBM could not be characterized by these data. Based upon the MetaboAnalyst pathway analysis results, seven major groups of altered metabolites differentiated TBM from other forms of meningitis (bacterial, viral, and fungal)—namely, amino acids, lipids, fatty acids, bilirubin, bile acids, and nucleosides. A more in-depth LC–MS metabolomics investigation was performed by van Laarhoven et al. [27] on an Indonesian adult (>18 years old) population of TBM cases. In an initial cohort of 33 TBM cases and 22 controls, 351 CSF metabolites were assessed—250 increased and 18 decreased in TBM [27]. Looking more closely at the TBM cases, 16 were identified as survivors (1 survivor was excluded from analysis due to aberrant LC–MS results) and 17 as non-survivors. Thirteen metabolites were identified as higher in survivors than in controls, and more elevated in non-survivors than in survivors. Of these 13 metabolites, eicosanoid leukotriene B4 was the metabolite that showed the greatest increase in non-survivors. Glucose and inositol revealed an opposite trend, being lower in survivors than in controls, and also lower in non-survivors than in survivors. Putrescine, cytidine, and tryptophan were found to be lower in non-survivors than in controls, and also lower in survivors than in non-survivors. Van Laarhoven et al. [27] identified cerebral tryptophan metabolism as one of the most upregulated metabolic pathways in TBM within this discovery cohort, with downstream metabolism in the kynurenine pathway being upregulated, but not associated with survival. Van Laarhoven et al. next examined CSF tryptophan in a validation cohort including 101 TBM cases and 17 controls, and identified indoleamine 2,3-dioxygenase—IDO 1 is an enzyme that catalyzes the first rate-limiting step in tryptophan catabolism—as showing greater expression in patients with TBM than in patients with brain trauma. Based upon both the discovery and validation cohorts, average CSF tryptophan concentrations for controls, non-survivors, and survivors were calculated to be 2.08 µmol/L, 1.11 µmol/L, and 0.2 µmol/L, respectively. Van Laarhoven et al. [27] went one step further with genotyping, and identified 11 independent, quantitative trait loci in 130 of the 133 TBM patients from both the discovery and validation cohorts. In an additional follow-up genetic validation cohort, consisting of 285 TBM cases (HIV negative), these 11 independent, quantitative trait loci, along with age and sex, were used to generate a composite prognostic index that strongly predicted survival among patients with TBM. Hence, van Laarhoven et al. [27] showed genetic association with cerebral tryptophan metabolism and mortality. There is, therefore, a causal role for tryptophan metabolism in outcomes of TBM. The strengths offered by these studies were the patient descriptions and follow-up, the combination of metabolomics and genomics results, and validation with two separate TBM patient cohorts. The limitations were that the discovery cohort was small, many variables in the sample were not annotated, and only HIV-negative cases were examined. Furthermore, van Laarhoven et al. [27] did not publish a full report of the metabolites characterizing TBM; hence, application to this review is limited.

In 2019, an untargeted NMR-based metabolomics study on CSF was conducted on an adult (15–70 years old) Chinese population, aiming to identify metabolic features and markers of TBM [29]. In the original cohort, 120 CSF samples were collected: TBM (*n* = 31), VM (*n* = 29), BM (*n* = 30), and controls (*n* = 30). After removal of outliers, 100 CSF samples—TBM (*n* = 25), VM (*n* = 27), BM (*n* = 20), and controls (*n* = 29)—were statistically analyzed. Zhang et al. [29] identified 21 CSF metabolites that characterized TBM. Increased lactate and decreased glucose were noted as the classic metabolic features of TBM, also related to the progression and prognosis of the disease. The other 19 CSF metabolites included increased alanine, isoleucine, valine, 3-hydroxybutyrate, 3-hydroxyisobutyrate, 2-hydroxyisovalerate, 2-oxoglutarate, caprate, isobutyrate, isovalerate, and acetamide; and decreased methionine, glycine, pyruvate, choline, myo-inositol, 1,3-dimethylurate, creatinine, and cyclohexane. Zhang et al. [29] also identified 23 and 6 CSF metabolites that distinguished TBM from VM and BM, respectively. The significance offered by the study of Zhang et al. [29] is that potential biomarkers for TBM were identified and, similar to Li et al. [21], the former study employed pathway analysis by MetaboAnalyst. In the discussion by Zhang et al. [29], some insights were offered that could shed more light on the metabolic characterization of TBM. Some limitations, also similar to Li et al. [21], were that only the 0.4–4.4 ppm NMR spectral region was analyzed, and the biological origin of cyclohexane could not be identified. Furthermore, as with the other TBM metabolomics studies, controls were cases suspected of having meningitis based on clinical symptoms, but later proved to be meningitis-negative, and the metabolic markers identified are putative to this cohort and require validation studies with larger cohorts to test their diagnostic value.

In a follow-up of their 2015 NMR metabolomics study, Mason et al. performed another metabolic characterization of a new South African TBM pediatric cohort (<12 years old) in 2020, again using untargeted NMR metabolomics [28]. From an original cache of 103 suspected TBM cases and 97 controls, a strict filtering of the samples was applied, and only 23 definite TBM cases and 33 controls were analyzed. The strength of this study by van Zyl et al. [28] was that stringent quality assurance steps were applied. Using 24 quality control (QC) samples, all NMR spectral regions that were consistently unreliable (coefficient of variance greater than 50%) were removed from analysis. Hence, only highly repeatable data were retained for statistical analysis. All cases (TBM and controls) with an HIV status that was positive or unknown were excluded. A screening of the metadata removed control cases that exhibited neurological symptoms or showed any signs of systemic TB, in order to obtain a homogeneous control group as close to normal as possible. For the experimental group, only cases with a final diagnosis of definite TBM [52] were included. The comparison of the “most normal” controls against the “most sick” cases of TBM allowed for one of the best metabolic characterizations of TBM in the literature. As with all of the other NMR-based CSF metabolomics studies, the two most discriminating CSF metabolites were reduced glucose and highly elevated lactate. Disregarding CSF glucose and lactate, TBM was characterized by 18 statistically significant metabolites (Table 1): 7 of these metabolites (alanine, choline, isoleucine, myo-inositol, pyruvate, and valine) corresponded with the study by Zhang et al. [29], and 10 metabolites (acetate, alanine, choline, citrate, creatinine, isoleucine, lysine, myo-inositol, pyruvate, and valine) overlapped with the previous study by Mason et al. [22]. Eight unique CSF metabolites (2-hydroxybutyrate, carnitine, creatine, creatine phosphate, glutamate, glutamine, guanidinoacetate, and proline) associated with TBM were identified for the first time by van Zyl et al. [28], and were linked to uncontrolled glucose metabolism, upregulated proline and creatine metabolism, detoxification, and the disrupted glutamate–glutamine cycle in TBM. Hence, these novel characteristic metabolites of TBM provided additional information towards our understanding of the condition. A hypothesis proffered by van Zyl et al. [28] was that increased levels of 2-hydroxybutyrate in the CSF of TBM cases supports the concept of insulin resistance and uncontrolled glucose metabolism in *M. tb-*infected brains. Uncontrolled glucose metabolism results in an inability of glucose to enter the cells, and subsequently leads to reduced cellular glycolysis and ATP deprivation, ultimately, leading to the destabilization of the blood–brain barrier. The major limitation of the report by van Zyl et al. [28] was that several unassigned variables contributed towards the discrimination of TBM from the control group—that is, there potentially remains some unexplored biological information in the CSF metabolic profiles of these TBM cases.

It should be noted that only van Laarhoven et al. [27] have validated their findings. All of the other CSF metabolomics studies on TBM reported here still need to be validated. Table 2 summarizes all of the CSF metabolomics studies in the current literature that focus on TBM in humans. It can be seen that the majority of CSF metabolomics studies are NMR-based, and that the number of TBM cases per study is relatively small, which is not surprising because TBM has a relatively low occurrence compared to TB.

Lastly, in a 1993 study pre-dating metabolomics [35], and a 2003 report that technically did not use the term “metabolomics” [36], Rodríguez-Núñez et al. adopted a targeted (metabolomics-like) LC–MS approach to identify adenosine monophosphate, guanosine, xanthine, and urate that characterized five pediatric TBM cases; and inosine, xanthine, and urate that characterized nine pediatric TBM cases. Rodríguez-Núñez et al. [35,36] are included in this review, as theirs are the only metabolic studies to report on nucleotides in TBM.

## 3. Metabolic Markers of TBM

The CSF contains a plethora of metabolic information that can be used to describe TBM. The two primary metabolic markers that consistently arise from CSF metabolomics studies of TBM are CSF lactate and CSF glucose—described below. These two primary metabolic markers of TBM often dominate the statistical results in NMR CSF metabolomics reports of the disease; hence, their removal is often required in order to assess the other metabolic markers of TBM. Table 1 summarizes the characteristic metabolites of TBM identified by CSF metabolomics studies. The remainder of this review will focus on giving biological context to the metabolic data in Table 1, by providing a holistic summary of the metabolic characteristics of TBM.

### 3.1. CSF Lactate

In the past, the presence of elevated CSF lactate in TBM cases was dismissed as being associated only with hypoxia. Indeed, it is true that lactic acid can be produced by anaerobic respiration—pyruvate is converted to lactate, catalyzed by lactate dehydrogenase and the oxidation of NADH to NAD^+^. However, the true metabolic role(s) of CSF lactate have just begun to emerge [24], with increasing evidence supporting the notion that CSF lactate has a novel, multifaceted perception in neurology [26]. One of the new roles proposed for CSF lactate in TBM cases is the astrocyte–microglia lactate shuttle (AMLS). The proposed shuttle implies that astrocytes produce elevated levels of lactate via glycolysis, and this lactate is transported via a shuttle system into the extracellular space and directed towards the microglia. The microglia take up the lactate, convert it to pyruvate, and use it immediately within the Krebs cycle and oxidative phosphorylation pathway for ATP production and the generation of free radicals, such as hydrogen peroxide, to aid in fighting the invading pathogen (*M. tb*).

#### Diagnostic Value

Numerous studies report CSF lactate to differentiate aseptic meningitis from bacterial meningitis. A meta-analysis of 33 studies (1974–2008) surmised a CSF lactate cutoff value of 3.885 mmol/L to differentiate the two forms of meningitis [53]. Another meta-analysis, of 25 studies (1974–2008), indicated that the CSF lactate cutoff value of reports differentiating aseptic meningitis from bacterial meningitis ranged from 2.1 to 4.44 mmol/L, with 12 studies having a CSF lactate cutoff value < 3.5 mmol/L, and 12 studies having a corresponding cutoff value ≥ 3.5 mmol/L [54]. While the exact cutoff value of CSF lactate in bacterial meningitis is still debated, it is clear that bacterial meningitis cases have abnormally high levels of CSF lactate. The information on CSF lactate levels in TBM in the literature is far more limited than in other forms of meningitis.

However, the presence of elevated CSF lactate in TBM is not new, having been identified as significantly elevated in early studies, such as by Donald and Malan in 1985 [37]. PubMed lists biochemical studies of CSF from TBM cases as far back as 1948, but these reports are either in an undetermined language (not in English) and/or not available online; hence, they are not included in this review. A clear cutoff value for CSF lactate in TBM to differentiate TBM from normal reference ranges or other forms of meningitis has not yet been determined in the literature. Table 3 lists 14 studies published between 1985 and 2020 that report CSF lactate values collectively from 462 TBM cases; from these, a weighted summary was calculated: TBM patients exhibit an average CSF lactate value of 6.59 ± 2.89 mmol/L, with a range of 3.04–17 mmol/L. The upper extent of the reference range of CSF lactate in pediatrics and adults is reported to be only 2.2 mmol/L and 2.7 mmol/L, respectively [50]. This indicates that even the lower extent of the range of CSF lactate in TBM (3.04 mmol/L) is greater than the upper extent of the reference range. Assigning a cutoff value for CSF lactate in TBM, however, is complicated by the fact that CSF lactate levels have been associated with the progression of TBM, with CSF lactate levels increasing as TBM becomes more severe, and associated with poor prognosis (indicated by a drop in the Glasgow coma score) [34]. Nonetheless, based on weighted calculations from existing studies, TBM patients tend to have higher levels of CSF lactate than subjects with bacterial meningitis, but determining a clear cutoff value of CSF lactate in TBM requires further research.

### 3.2. CSF Glucose

Reduced glucose concentrations and elevated levels of proteins in the CSF have long been the two biochemical indicators used to diagnose TBM. Table 3 shows that the weighted average of CSF glucose in cases of TBM is 1.88 ± 0.18 mmol/L, with a range of 1.6–2.69 mmol/L. These criteria are based on 11 studies, and collectively from 758 TBM patients (largely thanks to the 469 TBM cases reported by Solomons et al. [43]). The lower extent of the reference range of CSF glucose in pediatrics and adults is 1.9 mmol/L and 2.6 mmol/L, respectively [50]. Thus, there is overlap between low, but normal, CSF glucose levels, and abnormally low CSF glucose levels, in TBM. While CSF glucose has been the primary metabolite used in the differential diagnosis of TBM, the presence of “low” CSF glucose levels is simply not reliable enough, as there is no clear cutoff value, and “low” CSF glucose simply indicates increased glycolysis (that is, enhanced energy production). Comparing Table 3 and Table 1 shows that the CSF glucose values reported in these CSF metabolomics studies were higher (2.69 ± 1.22 mmol/L [22], 2.02 ± 0.79 mmol/L [29], and 2.38 ± 1.19 mmol/L [28]) than the calculated weighted average of 1.88 ± 0.18 mmol/L. Chatterji et al. [31] reported glucose as α-glucose (1.84 ± 1.27 mmol/L) and β-glucose (3.61 ± 3.06 mmol/L). Using the α:β ratio of 39:61 [55], an estimated average of total glucose for adult TBM in the 26 cases reported by Chatterji et al. [31] would be 2.92 mmol/L—still above the weighted average. The reasoning is unclear as to why the reported CSF glucose values in metabolomics studies are greater than in other reports. NMR is well known to be reliable and repeatable, and capable of producing highly accurate absolute quantitative data [56,57]; hence, the reason for the discrepancies is unlikely to be in the analytical instrument.

Based on the reported CSF lactate studies of TBM, there is a clear distinction between normal CSF lactate and abnormally high CSF lactate concentration values reported in cases of TBM. Hence, CSF lactate should be a more accurate indicator than glucose to diagnose TBM in the clinical setting.

### 3.3. Other Metabolic Markers of TBM

Only six CSF metabolomics studies have been reported in the literature (up to the end of 2020) that compared the CSF from TBM cases with controls [22,25,27,28,29,31], specifically identifying statistically significant metabolites that characterize TBM (Table 2). Five of these studies were NMR-based, whereas Mason et al. [25] conducted a GC–MS targeted study that reported only on amino acids. Three of these six CSF metabolomics reports involved adults (>18 years old)—the number of TBM cases used to characterize the disease were 27 [27], 26 [31], and 25 [29]. The other three studies involved pediatrics (<12 years old, but not newborns (<6 months)) in sample groups of 17 [22], 33 [25], and 23 [28] TBM cases.

The CSF metabolomics information reviewed here reveals that, CSF lactate and glucose aside, there are five main classes of metabolites that characterize TBM: amino acids, organic acids, nucleotides, carbohydrates, and “other” (Figure 3). Fifteen amino acids (alanine, asparagine, glutamate, glutamine, glycine, histidine, lysine, methionine, phenylalanine, proline, tryptophan, tyrosine, and the branched-chain amino acids—isoleucine, leucine, and valine) characterize TBM. Tryptophan and methionine have been reported to be reduced in TBM, and 12 amino acids are elevated in TBM cases, all compared to population control groups. All of these amino acid data are supported by one another across both adult and pediatric populations. The only discrepancy is that glycine was reported to be elevated in a South African pediatric TBM population [25] and lowered in a Chinese adult TBM population [29]. The carbohydrate myo-inositol has been uniformly reported as being decreased in TBM, with the increase in a byproduct of uncontrolled glucose metabolism—2-hydroxybutyrate. Four nucleotides characterizing TBM—decreased 1,3-dimethylurate, and increased inosine, xanthine, and urate—were reported by Rodríguez-Núñez et al. [35,36] using LC–MS. Five of the six CSF metabolomics studies reviewed here that characterize TBM were NMR-based. It should be noted that urate is invisible in a ^1^H-NMR spectrum, and 1 µmol/L is typically the limit of quantification in ^1^H-NMR metabolomics. Hence, none of these NMR-based studies would have confidently identified these nucleotides, except perhaps for xanthine. Eleven organic acids have been identified that characterize TBM: pyruvate, acetate, citrate, formate, 3-hydroxybutyrate, 3-hydroxyisovalerate, 2-hydroxyisovalerate, 2-oxogluturate, isobutyrate, isovalerate, and caprate. There are some discrepancies between studies with regard to whether these 11 organic acids are increased/decreased in TBM, when compared to their respective population control groups. However, the general trend is that most of these 11 organic acids are elevated compared to normal reference ranges. Lastly, nine “other” metabolites characterize TBM: choline, creatinine, dimethyl sulfone, acetamide, glycerophosphocholine, creatine, creatine phosphate, carnitine, and guanidinoacetate. These “other” metabolites, along with the four other classes of metabolites that characterize TBM, can all be linked to altered neuroenergetics [24]. The role of neuroenergetics during chronic neuroinflammatory conditions caused by infectious diseases such as TBM needs to be explored. Further examination of all of these CSF metabolites that characterize TBM should provide hypotheses that test our understanding of the disease and launch future research into TBM.

## 4. The Future of CSF Metabolomics in TBM Research

Each metabolomics analytical platform (NMR, GC–MS, LC–MS) provides a unique viewpoint of the CSF metabolome [49]. When used in combination (NMR + GC–MS, NMR + LC–MS, GC–MS + LC–MS), the output provides complementary data that support the metabolic interpretation(s) given. However, when all three metabolomics analytical platforms are used together, a more global view of the altered CSF metabolome induced by TBM is observable, providing unique insights. As the old idiom states, “the whole is greater than the sum of its parts”. Hence, it is our recommendation that a more global approach be taken in future CSF metabolomics studies of TBM, in order to provide more holistic hypotheses. Correlations of metabolic markers characterizing TBM should be investigated against indicators of neuroinflammation and/or TBM progression, such as the CSF:serum albumin ratio, which is an indicator of the disruption of the blood–brain barrier. Furthermore, incorporating other “omics” data, such as those from immunology, will be needed in order to predict and model the downstream metabolic effects of TBM [20].

It should also be noted that CSF metabolomics investigations of TBM are advancing from a proof-of-concept, exploratory phase that discovers new metabolites (in untargeted studies), towards an era in which new hypotheses, based on existing untargeted metabolomics studies, can be formulated, and mechanistic insights tested, in validation cohorts. Thereafter, metabolic markers as differential indicators of TBM can be tested and validated for earlier diagnosis and quicker treatment in order to curb this disease.

## Figures and Tables

**Figure 1 metabolites-11-00661-f001:**
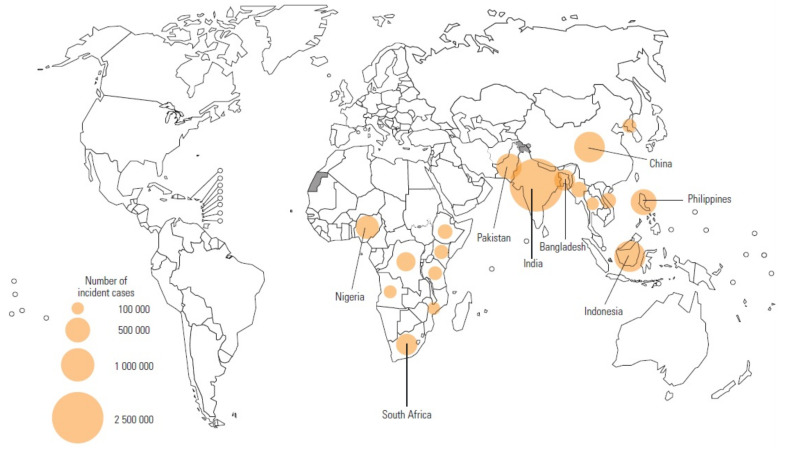
Geographical regions identified by the WHO as having the highest TB incidence in 2019. The top eight countries accounted for more than two-thirds of global cases: India (26%), Indonesia (8.5%), China (8.4%), the Philippines (6.0%), Pakistan (5.7%), Nigeria (4.4%), Bangladesh (3.6%), and South Africa (3.6%) [2].

**Figure 2 metabolites-11-00661-f002:**
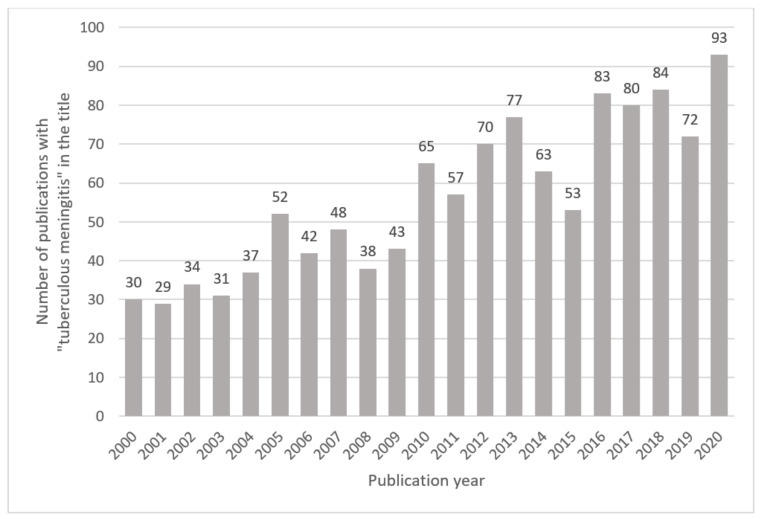
Number of publications with “tuberculous meningitis” in the title published between 2000 and 2020. Results are based on an online PubMed search.

**Figure 3 metabolites-11-00661-f003:**
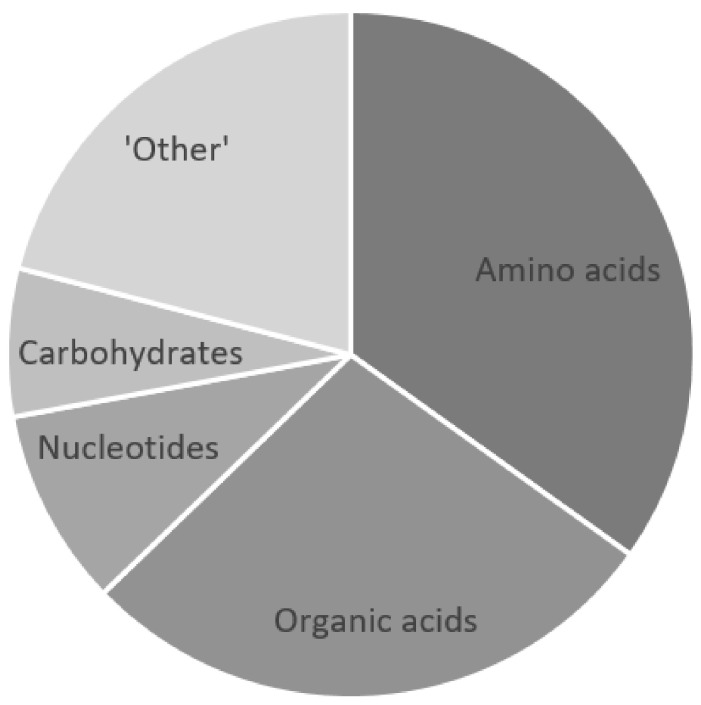
Five classes of metabolites that characterize TBM: amino acids, organic acids, nucleotides, carbohydrates, and “other”. The component “pie slices” quantitatively represent each metabolite class that contributes toward the metabolic characterization of TBM.

**Table 1 metabolites-11-00661-t001:** Summary of characteristic metabolites of TBM identified by CSF metabolomics studies. All concentrations given as µmol/L. All reference values, where available, were obtained from the Human Metabolome Database (www.hmdb.ca (accessed on 24 January 2021)) and/or Mandal et al. [49], except for glucose and lactate [50]. Red indicates elevated in TBM and blue indicates decreased in TBM. N/A = not available.

	Metabolites	Mason et al. 2015 [22]	Chatterji et al. 2016 [31]	Mason et al. 2017 [25]	Zhang et al. 2019 [29]	van Zyl et al. 2020 [28]	Rodríguez-Núñez et al. 2003 [36]	Reference Ranges	
TBM cases (*n*)		17 TBM	26 TBM	33 TBM	25 TBM	23 TBM	9 TBM		
Pediatric/Adult		Pediatric	Adult	Pediatric	Adult	Pediatric	Pediatric	Pediatric	Adult
Amino acids	Alanine	99.5 ± 71.9		77.3 ± 52	126.4 ± 82.2	154.7 ± 73.4		11–53.5	46 ± 27
	Asparagine			19.9 ± 14				4–10	4 ± 2
	Glutamate					54.4 ± 31.1		0–15	40 ± 52
	Glutamine					697.1 ± 213.9		285–820	398 ± 150
	Glycine			58.2 ± 31.7	6.3 ± 4.3			1–10	6.1 ± 1.4
	Histidine		18.69 ± 26.42					0–17	15 ± 8
	Isoleucine	54 ± 27.6	97.58 ± 438.33		55 ± 40.9	25.5 ± 18.6		1.2–7.5	7 ± 5
	Leucine							3.3–21.3	16 ± 9
	Lysine	86.6 ± 40.7		36.5 ± 25.2		183.5 ± 69.2		6–37	29 ± 13
	Methionine				5 ± 1.7			0–5	5 ± 4
	Phenylalanine	37.4 ± 18.4	72.04 ± 56.9					4–20	15 ± 8
	Proline			24.3 ± 25.8		105 ± 46.4		0–3	1.9 ± 1
	Tyrosine	31 ± 14.8	45.81 ± 32.56					3.9–18.5	12 ± 9
	Valine	69.5 ± 40.1	71.7 ±70		87.6 ± 56.6	71.5 ± 49.5		7.5–23	19 ± 13
Carbohydrates	Glucose	2689.1 ± 1217			2016.41 ± 792.65	2379.21 ± 1194.25		1900–4900	2600–4500
	α-Glucose		1841.14 ± 1271.65					N/A	N/A
	β-Glucose		3606.24 ± 3056.17					N/A	N/A
	myo-Inositol	177.3 ± 95.6	206.5 ± 163.87		79.35 ± 31.02	293.08 ± 181.89		N/A	84 ± 40
	2-Hydroxybutyrate		82.6 ± 115.26			108.22 ± 53.46		37 ± 21	40 ± 24
Organic acids	Lactate	7363.7 ± 2361.3	17019.32 ± 9554.84		7071.45 ± 3093.38	7593.41 ± 3592.63		1000–2200	1200–2700
	Pyruvate	82 ± 35.6			92.53 ± 28.78	88.8 ± 40.35		27 ± 11	53 ± 42
	Acetate	183.38 ± 111.22	6.16 ± 5.66			43.07 ± 15.74		284 ± 126	58 ± 27
	Citrate	206.02 ± 29.47				262.23 ± 53.72		N/A	255 ± 96
	Formate		8.04 ± 9.99					N/A	32 ± 16
	3-Hydroxybutyrate				126.89 ±110.95			N/A	34 ± 31
	3-Hydroxyisovalerate		2.15 ± 2.05		11.87 ± 9.26			N/A	4 ± 2
	2-Hydroxyisovalerate				15.78 ± 12.24			N/A	8 ± 6
	2-Oxogluturate				12.17 ± 5.56			2.1 ± 0.7	5 ± 4
	Isobutyrate		0.92 ± 0.57		7.32 ± 5.68			N/A	0–3.6
	Isovalerate				5.75 ± 3.35			N/A	0–2.7
	Caprate				12.86 ± 10.11			N/A	N/A
Nucleotides	1,3-Dimethylurate				1.87 ± 0.98			N/A	N/A
	Inosine						1.14 ± 1.02	0.0–1.2	0.64 ± 0.35
	Xanthine						6.14 ± 3.74	2.5 ± 0.7	13 ± 7
	Urate						88.87 ± 69.89	10.2 ± 6.8	15 ± 10
Other	Choline	4.3 ± 3.4			1.51 ± 1.03	5.31 ± 2.92		N/A	7–13
	Creatinine	47.7 ± 20.3			44.59 ± 19.81	78.55 ± 45.62		N/A	43 ± 12
	Dimethyl sulfone	4.9 ± 5.4						N/A	2 ± 1
	Acetamide				2.32 ± 1.84			N/A	N/A
	Glycerophosphocholine		0.78 ± 1.52					N/A	3.9 ± 1.2
	Creatine		8.31 ± 3.97			153.04 ± 57.5		N/A	44 ± 13
	Creatine phosphate					32.48 ± 18.11		N/A	N/A
	Carnitine					20.43 ± 8.55		N/A	1.9 ± 0.5
	Guanidinoacetate					165.16 ± 74.32		N/A	2.3 ± 0.9

**Table 2 metabolites-11-00661-t002:** List of CSF metabolomics studies from the literature focused on tuberculous meningitis in human participants.

Study	Pediatrics/ Adults	Population	TBM Cases (*n*)	Platform	Metabolites Characterizing TBM (*n*)
Coen et al. 2005 [30]	Adults	Australian	2	NMR	N/A
Mason et al. 2015 [22]	Pediatrics	South African	33	NMR	17
Chatterji et al. 2016 [31]	Adults	Indian	30	NMR	15
Li et al. 2017 [21]	Adults	Chinese	18	NMR	N/A
Mason et al. 2017 [25]	Pediatrics	South African	33	GC-MS	5
Dai et al. 2017 [19]	Adults	Chinese	50	LC-MS	N/A
van Laarhoven et al. 2018 [27]	Adults	Indonesian	32	LC-MS	6
Zhang et al. 2019 [29]	Adults	Chinese	31	NMR	21
van Zyl et al. 2020 [28]	Pediatrics	South African	23	NMR	20

N/A: not applicable; NMR: nuclear magnetic resonance spectroscopy; GC–MS: gas chromatography–mass spectrometry; LC–MS: liquid chromatography–mass spectrometry.

**Table 3 metabolites-11-00661-t003:** CSF lactate and CSF glucose concentrations reported in TBM, as described by 15 studies over the past 25 years, including their weighted summaries and comparison to reference values. Concentrations are in mmol/L. SD = standard deviation.

		Lactate				Glucose			
Study	Pediatrics/Adults	*n*	Mean	SD	Range	*n*	Mean	SD	Range
Donald and Malan 1985 [37]	Both	26	4.86			38	1.94		
Tang 1988 [38]	Adults	21	7	2.3	3.9–11	21	1.7	0.8	0.6–3.2
Donald et al. 1989 [39]									
Stage 2	Pediatrics	20	5.33		3.11–9.87	20	1.6		1–4
Stage 3	Pediatrics	14	5.77		2.53–6.95	14	1.6		1–5.2
Carlini et al. 1997 [33]	Pediatrics	61	6.4		1.4–17.9	79	1.8		0.1–9.1
Lu et al. 2001 [40]									
Poor outcome	Adults	16	7.16	2.165		18	1.78	0.966	
Good outcome	Adults	18	4.97	2.31		18	2.098	0.73	
Thwaites et al. 2002 [41]	Adults	102	5.4		1.5–9.8				
Thwaites et al. 2003 [42]									
Survived	Adults	16	5.9	2.6					
Died	Adults	5	9.5	3.3					
Mason et al. 2015 [22]	Pediatrics	17	7.36	2.36		17	2.69	1.22	
Solomons et al. 2015 [43]	Pediatrics					469	1.85	1.18	
Chatterji et al. 2016 [31]	Adults	26	17	9.55					
Mason et al. 2016 [23]	Pediatrics	20	5.2	2.6	1.5–10.5	16	1.96	1.35	0.2–5.1
Faried et al. 2018 [34]	Both	34	3.04	1.05	1.5–6.4				
Zhang et al. 2019 [29]	Adults	25	7.07	3.09		25	2.01	0.79	
Cresswell et al. 2020 [44]	Adults	18	9.5		4.6–11				
Van Zyl et al. 2020 [28]	Pediatrics	23	7.59	3.59		23	2.4	1.19	
**Weighted summary:**		**N = 462**	**6.59**	**2.89**	**3.04–17**	**N = 758**	**1.88**	**0.18**	**1.6–2.69**
Reference:									
Leen et al. 2012 [50]	Pediatrics				1.0–2.2				1.9–4.9
Leen et al. 2012 [50]	Adults				1.2–2.7				2.6–4.5

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
