# Peer review of "CSF Metabolomics of Tuberculous Meningitis: A Review"

_metabolites, 2021, doi:10.3390/metabo11100661_

Round 1

Reviewer 1 Report

"One of the first metabolomics studies involving CSF collected from humans with TBM was a 2005 pilot investigation.....". 
Is this statement true? May be it made sense to expand the search for other combinations of keywords, for example: (tuberculous meningitis) AND (gas chromatography); (tuberculous meningitis) AND (liquid chromatography); (tuberculous meningitis) AND (NMR).
The publication search in PubMed using keywords (tuberculous meningitis) AND (gas chromatography) resulted in 26 publications from 1977 year including, for example, the article doi10.1128/jcm.28.5.989-997.1990 about profile of carboxylic acids which are also low-molecularly weight metabolites and these articles should be discussed as metabolomic studies.
Also there are many articles about tuberculostearic acid which is also should be investigated.

Reviewer 2 Report

The authors provide a comprehensive review of CSF metabolomic (and metabolite) studies of TBM. Considering the growing importance of metabolomic insights and metabolite biomarkers for classification of CNS infections and neuroinflammatory diseases, this is a timely contribution of interest to both clinicians and basic researchers. It is a strength that the authors comment on both strengths and weaknesses of the included studies. Also, including lactate and glucose as separate entities in the review is of interest, as especially lactate is by now a very well validated CSF biomarker for bacterial meningitis. I only have minor suggestions for improvement.

It would be interesting to know whether any studies revealed correlations of any of the identified markers with the degree of neuroinflammation or disease activity. 

Can the authors provide a mechanistic hypothesis why lactate may tend to be higher in TBM than in other bacterial meningitis? 

It would be helpful to provide a better functional model hypothesizing why the metabolite classes listed in the pie chart are associated with TBM. 

It should be clarified early in the paper that the review does not cover lipidomics (not all readers, especially of clinical background are familiar with this a priori distinction).
